# Research on the Loran-C Pseudorange Positioning Method Based on an Ellipsoidal Geodesic Model and Its Application in Inland Areas

**DOI:** 10.3390/s25165110

**Published:** 2025-08-18

**Authors:** Ao Gao, Bing Ji, Miao Wu, Sisi Chang, Guang Zheng, Deying Yu, Wenkui Li

**Affiliations:** 1School of Electrical Engineering, Naval University of Engingeering, Wuhan 430000, China; 19815530099@139.com (A.G.); jibing1978@126.com (B.J.); roseoverthesea1130@126.com (S.C.); yudeying0716@126.com (D.Y.); li_wenkui123@163.com (W.L.); 292678troop, Tianjin 300000, China; a13582326212@126.com

**Keywords:** geodesy, LORAN, position, navigation, and timing (PNT)

## Abstract

The Loran-C system employs the spherical hyperbola positioning (SHP) method. However, SHP has three drawbacks in inland regions: first, approximating the Earth’s ellipsoid as a sphere introduces positioning errors; second, hyperbola positioning inherently suffers from a high geometric dilution of precision (GDOP) value; third, it is not easy to simultaneously receive long-wave signals from an entire chain of stations under complex propagation paths, which, to some extent, limits the application and development of the Loran-C system in inland areas. This paper addresses the limitations of the SHP algorithm and introduces the ellipsoidal pseudorange positioning (EPP) method, which eliminates the need to approximate the Earth’s ellipsoid as a sphere. This pseudorange positioning algorithm reduces the GDOP value, enabling navigation and positioning with signals from just three stations, thereby breaking through the restriction of requiring signals from a single chain. Simulation analyses were conducted for various Loran-C chains in China. Due to differences in the geometric layout of the chains, the EPP algorithm improved the positioning coverage area by 129.1% to 284.6% compared to the SHP algorithm. In field test data from the Maoming region of Guangdong Province, China (a typical inland mountainous environment), the EPP algorithm significantly reduced the root mean square error (RMSE), from 417.2 m with the SHP algorithm to 43.1 m, representing an improvement of 89.7%. Both the simulation and experimental results demonstrate that the EPP method effectively addresses errors in Earth ellipsoid modeling, significantly reduces the GDOP, and substantially improves the positioning accuracy and usable area of the Loran-C system in complex inland terrain. This provides more reliable technical support for Loran-C applications in inland navigation, timing, and timing backup.

## 1. Introduction

The Loran-C system is a medium- and long-range precision radio navigation system; the basic working principle is to receive the same station chain at a certain point of the primary and secondary station signal arrival time difference. Through the stability of the propagation speed of the radio wave, the time difference is converted into a distance difference. The trajectory of the point with the same distance difference is a hyperbola with the transmitter as the focus, and the intersection of the two sets of hyperbolas can be used to determine the positioning point [1,2,3]. Using the same group of stations to locate the chain is called single-station chain positioning. The accuracy of this positioning method is limited by the way that the stations are laid out, and the best positioning area is located in the fan-shaped area on one side of the positive baseline [4,5]. During the single-station chain positioning process, singular phenomena may occur near the station chain baseline, making it impossible to locate [6,7] and further restricting the occasions wherein single-station chain positioning can be used. Existing Loran-C devices using single-station chain positioning can achieve low positioning accuracy of 0.25 nautical miles (460 m) with timing accuracy of 0.5–1 μs at 95% availability. Based on the heavy constraints of Loran-C single-station chain positioning, concerning the positioning method of the GNSS system, which is also a radio navigation system, a pseudoranging positioning method can be used to obtain the user’s spatial coordinates. Moreover, more multi-dimensional information can be obtained through the introduction of at least three transmitter stations, which can be used to derive the user’s elevation. Many researchers have recently transplanted this positioning method to the Loran-C system, thus obtaining a pseudorange positioning method applicable to the eLoran system [8].

Although Loran-C involves two-dimensional positioning, due to the existence of the clock difference, to obtain the user’s position information, at least three stations need to participate in the positioning at the same time to obtain three-dimensional information (one more dimension is the time difference). In the traditional hyperbolic method for cross-station chain positioning, there are certain limitations, e.g., there must be two or two stations belonging to the same station chain. The pseudorange positioning method [9] does not need the TD between known stations to obtain the positioning results. The TOAs of stations belonging to different station chains can also be used to obtain the positioning results, which significantly simplifies the cross-station chain positioning method of the Loran-C system, breaks through the limitations of the traditional hyperbolic method on the same station chain, and further promotes the positioning accuracy and usable range of the Loran-C system [10].

Jinghu et al. [11] and Fengchang et al. [12] derived a pseudorange localization algorithm for the spherical plane based on approximating the Earth’s ellipsoid as a sphere. However, since the Earth more closely approximates a rotating ellipsoid, an error will arise between the results obtained using a sphere [13] and the actual value. When the sphere is established with a long semi-axis, the error will be further enlarged at high latitudes, thus weakening the advantage of pseudorange positioning over traditional single-station chain positioning.

Son W P et al. [14] discussed the application of eLoran near ports. Korea’s eLoran system can calculate the absolute position with accuracy of approximately 15 m with 95% probability at the port approach stage. However, for inland complex terrain, the lack of an eLoran station with ASF information support significantly reduces the positioning accuracy, making it especially important to improve the algorithm’s accuracy.

Based on the limitations and shortcomings of single-station chain positioning, this paper introduces a spherical, hyperbolic positioning algorithm that utilizes the spherical angular distance and the spherical latitude/longitude derivative problem. It derives a pseudorange positioning formula based on the geodesic length and the geodesic latitude derivative to obtain a pseudorange positioning method on the ellipsoid, which improves the positioning accuracy of the positioning method itself to a certain extent and eliminates the singularity phenomenon.

The system characteristics of Loran-C indicate that its positioning accuracy is significantly lower than that of GNSS. Loran-C generally serves as a backup for GNSS, and this paper aims to discuss the use of Loran-C in scenarios where it is not integrated with GNSS. Based on the limitations and shortcomings of single-station chain localization, this paper introduces a spherical hyperbola localization algorithm that addresses the issues of spherical angular distance and spherical latitude/longitude derivatives. It derives the pseudodistance localization formula based on the geodesic length and the geodesic latitude derivative, obtaining a pseudorange localization method for the ellipsoid surface, which improves the positioning accuracy of the localization method itself and eliminates the singularity phenomenon.

Regarding the problem whereby some regions cannot receive signals from all three transmitters in a group of station chains simultaneously, but can receive signals from multiple stations of different station chains, cross-station chain positioning is required to solve the problem of calculating the cross-station chain clock difference. In this paper, we posit that accessing the GNSS time reference can effectively solve the cross-station chain clock difference problem; however, this is not consistent with the use of Loran-C as a GNSS backup system. In this paper, the Extended Kalman Filter (EKF) is employed to estimate the cross-station chain clock difference, which in turn enables cross-station chain pseudorange localization and expands the application scope of Loran-C.

In Section 2 of this paper, ground wave transmission theory and the Loran-C pseudorange localization algorithm will be introduced; in Section 3, the cross-station chain clock difference estimation method based on the EKF will be introduced; in Section 4, simulations and experiments will be carried out; and, in Section 5, the conclusions of the article will be given.

## 2. Geodesic Issues in eLoran System Positional Solving

Loran-C navigation involves two key components: ground wave propagation and positioning calculation. This section primarily introduces the theory of ground wave propagation and the calculation methods for positioning. Section 2.1, Section 2.2 and Section 2.3 briefly describe the theory of ground wave propagation, providing a theoretical basis for subsequent SF+ASF calculations. Section 2.4, Section 2.5 and Section 2.6 derive the ellipsoidal pseudorange positioning formula, improving the positioning accuracy from an algorithmic perspective.

### 2.1. Ground Wave Transmission Theory

According to ground wave propagation theory, the time of arrival (*TOA*) of Loran-C radio waves transmitted between two points can be expressed as(1)$$\begin{array}{c} TOA=PF+SF+ASF \end{array}$$

According to electromagnetic wave transmission theory, for any ground wave path, the vertical component of the signal electric field is [15](2)$$\begin{array}{c} E_{r}\left(\mu V/m\right)=j\frac{3\times 10^{5}}{d}\sqrt{P}\times \left|W_{g}\right|\times e^{j(\omega t-n_{s}\times \frac{d}{c}-\arg W_{g})} \end{array}$$

In the formula, *j*: Plural unit;*P*: Radiation power, in kW;*d*: Distance between the two locations, measured along the ground, in kilometers;$n_{s}$: Atmospheric refractive index along the propagation path on the Earth’s surface;ω: ω=2πf ($f=10^{5}$ Hz radio frequency);*c*: Speed of light in a vacuum;$W_{g}$: Ground wave attenuation factor, which is a complex function of the equivalent conductivity of the ground along the propagation path, the relative permittivity, the atmospheric refractive index and its gradient, and the path distance *d*;$\left|W_{g}\right|$: Modulus of the ground wave attenuation factor;$\arg W_{g}$: Beam angle of the ground wave attenuation factor.

From Equation (Equation 2), we can obtain the formula for calculating the ground wave transmission delay:(3)$$\begin{array}{c} TOA=t_{0}+t_{w} \end{array}$$(4)$$\begin{array}{c} t_{0}=\frac{n_{s}d}{c}\times 10^{6} \end{array}$$(5)$$\begin{array}{c} t_{w}=\arg\frac{W_{g}}{\omega }\times 10^{6} \end{array}$$

Among these, $t_{0}$ is referred to as the basic delay (also known as the primary phase factor), which is the time taken for a signal to propagate from the transmitting antenna to the receiving antenna in an infinite uniform air medium, i.e., PF. Due to differences in atmospheric environmental characteristics across regions, the USCG specified in 1981 [16] that $n_{s}=1.000338$ should be selected, with *c* = 299,792,458 m/s corresponding to it and *d* representing the geodesic distance between two points. $t_{w}$ is referred to as the secondary delay (also known as the secondary phase factor), which is the difference between the propagation delay and the primary delay, reflecting the influence of seawater and the ground on the propagation delay [17]. $t_{w}$ is the sum of SF and ASF. The experiments in this paper were primarily conducted in inland areas, where Loran-C signals rarely propagate over the sea surface. Using $t_{w}$ to calculate the delay simplifies the calculation process and improves the computational efficiency. Thus, calculating the TOA is converted into calculating the ground wave attenuation factor $W_{g}$.

### 2.2. Diffraction Formula for Smooth Spherical Ground Waves

The wavelength λ=3000 m of 100 kHz electromagnetic waves. For obstacles with ground elevation variations of less than 500 m, the ground can be regarded as a smooth spherical surface. For 100 kHz electromagnetic waves, the boundary between spherical waves and plane waves is 70 km. Loran-C is a long-distance navigation tool, and the distance between the station and the observation point is generally several hundred kilometers. At this point, the ground wave attenuation factor $W_{g}$ can be expressed as(6)$$\begin{array}{c} W_{g}=e^{j\frac{\pi }{4}}\sqrt{\pi x}\underset{s=1}{\sum^{\infty }}\;e^{jxt_{s}}\frac{w\left(t_{s}-y_{1}\right)w\left(t_{s}-y_{2}\right)}{\left(t_{s}-q^{2}\right){\left[w(t_{s})\right]}^{2}} \end{array}$$

In the formula,(7)$$\begin{array}{c} x={\left(\frac{Ka_{e}}{2}\right)}^{\frac{1}{3}}\frac{10^{3}d}{a_{e}} \end{array}$$(8)$$\begin{array}{c} y_{i}={\left(\frac{Ka_{e}}{2}\right)}^{-\frac{1}{3}}kh_{i}\;\left(i=1,2\right) \end{array}$$(9)$$\begin{array}{c} q=j{\left(\frac{Ka_{e}}{2}\right)}^{\frac{1}{3}}\frac{\sqrt{\varepsilon -1+j60\lambda \Sigma }}{\varepsilon +j60\lambda \Sigma } \end{array}$$(10)$$\begin{array}{c} K=2\pi /\lambda \end{array}$$

Among these, $a_{e}$ is the equivalent Earth radius under standard atmospheric conditions; $a_{e}=\frac{4}{3}r$, where *r* is the actual Earth radius, with a value of 6,378,137 m; $h_{1},h_{2}$ are the heights of the transmit and receive antennas above the ground, respectively; λ is the wavelength of electromagnetic waves in air; K=2π/λ; σ is the equivalent electrical conductivity of the Earth, with a unit of S/m; ε is the relative permittivity of the Earth; σ,ε can be obtained by consulting reference [17]. $w(t_{s})$ is the Airy function. When $h_{1},h_{2}\to 0$, $y_{1},y_{2}\to 0$,(11)$$\begin{array}{c} w\left(t_{s}-y_{1}\right)=w\left(t_{s}-y_{2}\right)=w\left(t_{s}\right) \end{array}$$

At this point, $W_{g}$ can be simplified to(12)$$\begin{array}{c} W_{g}=e^{j\frac{\pi }{4}}\sqrt{\pi x}\underset{s=1}{\sum^{\infty }}\;e^{jxt_{s}}\frac{1}{t_{s}-q^{2}} \end{array}$$

In Equation (Equation 12), $t_{s}$ is the s-th complex root of the differential equation $\frac{dt}{dq}=\frac{1}{t-q^{2}}$. $t_{s}$ is calculated using the classical Runge–Kutta method, as described in reference [18].

The calculation of $W_{g}$ requires the key parameters Σ, ε according to reference [17], as shown in Table 1.

This section mainly completes the correction of the TOA of the Loran-C receiver, i.e., by locating the approximate position, reverse-calculating the ground wave transmission path, calculating the delay TOA caused by ground wave transmission, and correcting *d*, thereby providing a more accurate distance for pseudorange positioning.

### 2.3. Method for Calculating Quadratic Delay Values on Segmented Uneven Smooth Path

In a segmented non-uniform smooth spherical ground model, the propagation path from the transmission point to the reception point is divided into finite segments based on the characteristics of the terrain along the radio wave transmission path, and the equivalent conductivity of each segment of the ground is constant. For segmented non-uniform smooth path models, the Millington formula [18] is commonly used in engineering.

As shown in Figure 1, suppose that the path is divided into segments, with conductivities $\Sigma_{1},\Sigma_{2},\Sigma_{3},\ldots ,\Sigma_{n}$ and corresponding geodesic distances $d_{1},d_{2},d_{3},\ldots ,d_{n}$. For a mixed path consisting of segments, the ground wave attenuation factor can be expressed as shown below:(13)$$\begin{array}{c} W_{g}=\sqrt{W_{p}W_{n}} \end{array}$$

In the formula,(14)$$\begin{array}{c} W=\frac{W\left(d_{1},\Sigma_{1}\right)W\left(d_{1}+d_{2},\Sigma_{2}\right).........W\left(d_{1}+d_{2}+...+d_{n},\Sigma_{n}\right)}{W\left(d_{1},\Sigma_{2}\right)W\left(d_{1}+d_{2},\Sigma_{3}\right).........W\left(d_{1}+d_{2}+....d_{n-1},\Sigma_{n}\right)} \end{array}$$(15)$$\begin{array}{c} W=\frac{W\left(d_{n},\Sigma_{n}\right)W\left(d_{n}+d_{n-1},\Sigma_{n-1}\right).........W\left(d_{n}+d_{n-1}+....+d_{1},\Sigma_{1}\right)}{W\left(d_{n},\Sigma_{n-1}\right)W\left(d_{n}+d_{n-1},\Sigma_{n-2}\right).........W\left(d_{n}+d_{n-1}+....+d_{2},\Sigma_{1}\right)} \end{array}$$

The advantage of the Millington method is that it is practical and straightforward. The disadvantage is that it is only applicable when both the transmitting and receiving points are on the ground. If the receiving point is in the air, it cannot be calculated. Loran users are generally on the ground and can use the Millington formula. From Equation (Equation 16), SF+ASF can be calculated as(16)$$\begin{array}{c} {t_{SF+}}_{ASF}=\frac{1}{2}\left(t_{p}+t_{n}\right) \end{array}$$(17)$$\begin{array}{c} t_{p}=t_{1}\left(\sigma_{1},d_{1}\right)+\underset{i=2}{\sum^{n}}\;\left[t_{i}\left(\sigma_{i},\underset{j=1}{\sum^{i}}\;d_{j}\right)-t_{i}\left(\sigma_{i},\underset{j=1}{\sum^{i-1}}\;d_{j}\right)\right] \end{array}$$(18)$$\begin{array}{c} t_{n}=t_{n}\left(\sigma_{n},d_{n}\right)+\underset{i=1}{\sum^{n-1}}\;\left[t_{n-i}\left(\sigma_{n-i},\underset{j=0}{\sum^{i}}\;d_{n-j}\right)-t_{n-i}\left(\sigma_{n-i},\underset{j=0}{\sum^{i-1}}\;d_{n-j}\right)\right] \end{array}$$

In Equations (Equation 17) and (Equation 18), $\Sigma_{i}$ is the Earth’s conductivity, and $\sum_{i=2}^{n}$, which also contains the upper and lower corner symbols, is the cumulative symbol.

### 2.4. Equation for Calculation of Earth Line Length

The geodesic length is the shortest distance between two points on the reference ellipsoid of the Earth. To simulate the ideal path for Loran signal propagation, calculating the geodesic length is essential in the latitude and longitude transformation of the eLoran system [19,20]. There are many formulas for the calculation of geodesics, among which the Andoyer–Lambert formula is simple, compact, symmetric, and fast, enabling real-time calculation. It is the earliest and the most widely used method in the field of navigation, with only a few meters of error in the range of 6000 km. Its accuracy meets the needs of Loran system positioning [21], and it is also the main formula used in applying and deducing the change in the latitude and longitude of the Loran system. The Andoyer–Lambert formula is derived based on the Bessel projection condition, in which distance calculation and azimuth calculation are separated. *S* is the geodesic length, σ is the spherical angular distance, *a*, *b* are the semi-major axis and semi-minor axis, *e* is the first eccentricity of the ellipsoid, *B*, *L* are the geodesic latitude and longitude, and φ, λ are the normalized latitude and longitude. We have(19)$$\left\{\begin{array}{cc} \tan\varphi & =\frac{b}{a}\tan B\\ \lambda & =L \end{array}\right.$$

The Bessel projection method involves projecting a projection on an ellipsoid onto a reference sphere. It satisfies the following conditions:(1)The spherical latitude of a point on the ellipsoid after projection to the sphere is equal to the naturalized latitude of the origin of the point of projection;(2)The geodesic on the ellipsoid is projected as a great arc on the sphere, and the spherical azimuth between the great arc and the spherical meridian is equal to the geodesic azimuth of the corresponding point.

Using the above conditions, we derive the differential formula for the calculation of the arc length on an ellipsoid surface [20]:(20)$$\left\{\begin{array}{cc} \Delta S & =a\sqrt{1-e^{2}\cos^{2}\varphi }\cdot \Delta \sigma \\ \Delta L & =\sqrt{1-e^{2}\cos^{2}\varphi }\cdot \Delta \lambda \end{array}\right.$$

Δσ is the spherical angular distance differential increment, and Δλ is the longitude difference differential increment, taking into account $e^{2}=2f-f^{2}=2f$ (*f* is the ellipsoidal oblateness):(21)$$sin\varphi =\cos A_{0}\sin\sigma$$

$A_{0}$ is the azimuth of the geodesic in the equatorial plane that exists:(22)$$sinA_{0}=\frac{\cos B_{1}\sin A_{1}}{\sqrt{1-e^{2}\sin^{2}B_{1}}}$$

The ΔS integrals can be obtained as follows:(23)$$\begin{array}{cc} \; & \Delta S=a\sqrt{1-2f\cos^{2}\varphi }\cdot \Delta \sigma \\ \; & =a\left(1-f\cos^{2}\varphi \right)\Delta \sigma \\ \; & =a\left(1-f+f\cos^{2}A_{0}\frac{1-\cos2\sigma }{2}\right)\Delta \sigma \end{array}$$

The ΔS integrals can be obtained as follows:(24)$$\begin{array}{c} S=a\left[\Delta \sigma \left(1-f+\frac{f}{2}\cos^{2}A_{0}\right)-\frac{f}{4}\cos^{2}A_{0}\left(\sin2\sigma_{2}-\sin2\sigma_{1}\right)\right] \end{array}$$
where $\Delta \sigma =\sigma_{2}-\sigma_{1}$, and, due to Equation (Equation 21), we have(25)$$\begin{array}{c} \sin\varphi_{1}+\sin\varphi_{2}=\cos A_{0}\left(\sin\sigma_{1}+\sin\sigma_{2}\right)=2\cos A_{0}\sin\frac{\sigma_{1}+\sigma_{2}}{2}\cos\frac{\Delta \sigma }{2} \end{array}$$(26)$$\begin{array}{c} \sin\varphi_{1}-\sin\varphi_{2}=\cos A_{0}\left(\sin\sigma_{1}-\sin\sigma_{2}\right)=2\cos A_{0}\cos\frac{\sigma_{1}+\sigma_{2}}{2}\sin\frac{\Delta \sigma }{2} \end{array}$$

This can be obtained from the trigonometric relationship(27)$$\begin{array}{c} U={\left(\sin\varphi_{1}+\sin\varphi_{2}\right)}^{2}=2\cos^{2}A_{0}\sin^{2}\frac{\sigma_{1}+\sigma_{2}}{2}\left(1+\cos\Delta \sigma \right) \end{array}$$(28)$$\begin{array}{c} V={\left(\sin\varphi_{1}-\sin\varphi_{2}\right)}^{2}=2\cos^{2}A_{0}\cos^{2}\frac{\sigma_{1}+\sigma_{2}}{2}\left(1-\cos\Delta \sigma \right) \end{array}$$

Substituting Equations (Equation 27) and (Equation 28) into Equation (Equation 24) and collating them gives(29)$$\begin{array}{cc} \; & \frac{S}{a}=\Delta \sigma \left\{1-f+\frac{f}{2}\left[\frac{U}{2(1+\cos\Delta \sigma )}+\frac{V}{2(1-\cos\Delta \sigma )}\right]\right\}\\ \; & -\frac{f}{2}\sin\Delta \sigma \left[\frac{V}{2(1-\cos\Delta \sigma )}-\frac{U}{2(1+\cos\Delta \sigma )}\right] \end{array}$$

Δσ can be calculated using the following equation:(30)$$\begin{array}{c} \cos\Delta \sigma =\sin\varphi_{1}\sin\varphi_{2}+\cos\varphi_{1}\cos\varphi_{2}\cos\Delta \lambda \end{array}$$

Since Δλ is unknown, $\Delta L=L_{2}-L_{1}$ is used instead:(31)$$\begin{array}{c} \cos\Delta \sigma^{\prime }=\sin\varphi_{1}\sin\varphi_{2}+\cos\varphi_{1}\cos\varphi_{2}\cos\Delta L \end{array}$$

Differentiating the above equation gives [20](32)$$\begin{array}{cc} \; & d\Delta \sigma =\frac{\cos\varphi_{1}\cos\varphi_{2}\sin\Delta L}{\sin\Delta \sigma }\cdot d\Delta L\\ \; & =\cos\varphi_{2}\frac{\sin(2\pi -A_{2})}{\sin\Delta \sigma }\sin\Delta L\cdot d\Delta L\\ \; & =-\cos\varphi_{2}\sin A_{2}d\Delta L\\ \; & =\sin A_{0}d\Delta L \end{array}$$

Then, we have(33)$$\begin{array}{c} \Delta L=\Delta \lambda -f\Delta \sigma \sin A_{0} \end{array}$$

Therefore,(34)$$\begin{array}{c} d\Delta L=\Delta \lambda -\Delta L=f\Delta \sigma \sin A_{0} \end{array}$$

Substituting this into Equation (Equation 32) gives(35)$$\begin{array}{c} d\Delta \sigma =\sin A_{0}\cdot d\Delta L=f\Delta \sigma \sin^{2}A_{0} \end{array}$$

Deformation leads to(36)$$\begin{array}{c} \Delta \sigma =\Delta \sigma^{\prime }+d\Delta \sigma =\Delta \sigma^{\prime }+f\Delta \sigma^{\prime }\sin^{2}A_{0}=\Delta \sigma^{\prime }\left(1+f\sin^{2}A_{0}\right) \end{array}$$

Finally,(37)$$\begin{array}{c} S=a\Delta \sigma^{\prime }-\frac{f}{4}a\left[\frac{\Delta \sigma^{\prime }-\sin\Delta \sigma^{\prime }}{1+\cos\Delta \sigma^{\prime }}U+\frac{\Delta \sigma^{\prime }+\sin\Delta \sigma^{\prime }}{1-\cos\Delta \sigma^{\prime }}V\right] \end{array}$$

The Andoyer–Lambert formulae are used several times to eliminate tiny quantities $f^{2}$. Finally, formulae suitable for navigation applications with a concise shape and symmetrically compact operations are obtained. These formulae are accurate to the flat-rate level and can meet the accuracy requirements of the eLoran system.

### 2.5. Derivative Problem Related to Geodesic Line Length and Geodesic Latitude and Longitude

The relationship between the change ΔS in the geodesic length on the rotating ellipsoid and the changes ΔB and ΔL in the latitude and longitude of the geodesic coordinates of a moving point $\left(B,L\right)$ on the rotating ellipsoid is [20](38)$$\begin{array}{c} \Delta S=-M\cdot \cos A\cdot \Delta B-N\cdot \cos B\cdot \sin A\cdot \Delta L \end{array}$$(39)$$\begin{array}{c} \left\{\begin{array}{c} \frac{𝜕S}{𝜕B}=-M\cos A\\ \frac{𝜕S}{𝜕L}=-N\cos B\sin A \end{array}\right. \end{array}$$
where *M* is the radius of curvature of the meridian circle,(40)M=a1−e2/1−e2sin2B32
where *N* is the radius of curvature of the prime vertical circle,(41)$$\begin{array}{c} N=a/\sqrt{1-e^{2}\sin^{2}B} \end{array}$$
where *B* and *L*, respectively, are the geodesic latitude and geodesic longitude; *a* is the rotation of the ellipsoid’s long half-axis length; $e^{2}$ is the first eccentricity of the square; *A* is the the azimuth of the geodesic line from the moving point to the fixed point $\left(B_{1},L_{1}\right)$; and the change in the relationship is as shown in Figure 2.

Due to the low signal-to-noise ratio at some Loran signal reception points and the projected spherical coordinate transformation, convergence to the farther point of the deviation after iteration is easy, leading to the singularity phenomenon when using this method. To obtain the geodesic lengths *S* of the geodesic latitude *B* and geodesic longitude *L* accurately, the derivation and simplification of the Andoyer–Lambert Equation (Equation 37) can be used to obtain the corresponding 𝜕S/𝜕B and 𝜕S/𝜕L:(42)$$\left\{\begin{array}{c} \frac{𝜕S}{𝜕B}=\frac{a}{8}e^{2}[\sigma \left(-U+V+\left(U+V\right)\cos\sigma \right)\cdot {\left(\csc\sigma \right)}^{4}\\ \times \left(-E-F\left(U+V\right)+F\left(U-V\right)\cos\sigma +E\cos2\sigma \right)]\\ -\frac{a\left(4F+e^{2}E\left(U+V\right)-e^{2}E\left(U-V\right)\cos\sigma \right)}{4\sin\sigma }\\ \frac{𝜕S}{𝜕L}=-\frac{aWe^{2}\sigma }{16}\left[-4\left(U^{2}+V^{2}\right)\cos\sigma +\left(U^{2}-V^{2}\right)\left(3+\cos2\sigma \right)\right]{\left(\csc\sigma \right)}^{4}\\ +\frac{aW}{\sin\sigma } \end{array}\right.$$

The coefficients in Equation (Equation 42) are(43)U=1−e2tanB1+1−e2tan2B+1−e2tanBi1+1−e2tan2BiV=−1−e2tanB1+1−e2tan2B+1−e2tanBi1+1−e2tan2BiE=−1−e2sec2B1+1−e2tan2B32F=e2−1sec2Bcosλ−λitanB−tanBi1+1−e2tan2B321+1−e2tan2BiW=sinλ−λi1+1−e2tan2B1+1−e2tan2Bi

### 2.6. Pseudorange Localization Methods for Approximate Spherical Projections

The spherical pseudorange positioning algorithm converts the TOA of each station that the Loran-C receiver receives into a distance. However, considering that there is a clock difference between the signal transmission time between the stations and the station’s clock, and there is also a time deviation between the receiver and the transmitter, the system of positioning equations based on the pseudorange can be expressed as(44)$$\begin{array}{c} D_{i}=S_{i}-t_{i}\cdot V_{0},\left(i=1,..,n\right) \end{array}$$
where the pseudorange observation is obtained by the receiver after deducting the transmitter broadcast control error and receiver system delay $D_{i}=TOA_{i}\cdot V_{0}$, and the received TOA of the receiver is converted into the pseudorange length, $S_{i}$, which is the geodesic length. After linearization using Newton’s iterative method and its Taylor expansion, retaining the primary term, we obtain the second relationship equation k−1:(45)$$\left\{\begin{array}{c} D_{1}-S_{1,k-1}=\frac{𝜕S_{1,k-1}}{𝜕B}\Delta B+\frac{𝜕S_{1,k-1}}{𝜕L}\Delta L+\frac{𝜕S_{1,k-1}}{𝜕t}\Delta t\cdot V_{0}+\varepsilon_{1}\\ D_{2}-S_{2,k-1}=\frac{𝜕S_{2,k-1}}{𝜕B}\Delta B+\frac{𝜕S_{2,k-1}}{𝜕L}\Delta L+\frac{𝜕S_{2,k-1}}{𝜕t}\Delta t\cdot V_{0}+\varepsilon_{2}\\ \begin{array}{cccc} & & & \ldots \end{array}\\ D_{n}-S_{n,k-1}=\frac{𝜕S_{n,k-1}}{𝜕B}\Delta B+\frac{𝜕S_{n,k-1}}{𝜕L}\Delta L+\frac{𝜕S_{n,k-1}}{𝜕t}\Delta t\cdot V_{0}+\varepsilon_{n} \end{array}\right.$$

Writing it in matrix form, we have(46)$$\begin{array}{c} G\cdot \Delta x+V=Z \end{array}$$(47)$$G=\left[\begin{array}{ccc} \frac{𝜕S_{1,k-1}}{𝜕B} & \frac{𝜕S_{1,k-1}}{𝜕L} & \frac{𝜕S_{1,k-1}}{𝜕t}V_{0}\\ \frac{𝜕S_{2,k-1}}{𝜕B} & \frac{𝜕S_{2,k-1}}{𝜕L} & \frac{𝜕S_{2,k-1}}{𝜕t}V_{0}\\ & \ldots & \\ \frac{𝜕S_{n,k-1}}{𝜕B} & \frac{𝜕S_{n,k-1}}{𝜕L} & \frac{𝜕S_{n,k-1}}{𝜕t}V_{0} \end{array}\right]$$(48)$$\begin{array}{c} \Delta x=\left[\begin{array}{c} \Delta B\\ \Delta L\\ \Delta t \end{array}\right] \end{array}$$(49)$$\begin{array}{c} V=\left[\begin{array}{c} \varepsilon_{1}\\ \varepsilon_{2}\\ \vdots \\ \varepsilon_{n} \end{array}\right] \end{array}$$(50)$$\begin{array}{c} Z=\left[\begin{array}{c} D_{1}-S_{1,k-1}\\ D_{2}-S_{2,k-1}\\ \vdots \\ D_{n}-S_{n,k-1} \end{array}\right] \end{array}$$

Since it is independent of the time change, Equation (Equation 47) can be transformed to(51)$$\begin{array}{c} G=\left[\begin{array}{ccc} \frac{𝜕S_{1,k-1}}{𝜕B} & \frac{𝜕S_{1,k-1}}{𝜕L} & V_{0}\\ \frac{𝜕S_{2,k-1}}{𝜕B} & \frac{𝜕S_{2,k-1}}{𝜕L} & V_{0}\\ & \ldots & \\ \frac{𝜕S_{n,k-1}}{𝜕B} & \frac{𝜕S_{n,k-1}}{𝜕L} & V_{0} \end{array}\right] \end{array}$$

Existing Loran positioning solution methods mostly approximate the Earth as a sphere, employ hyperbolic or circular positioning, and utilize the TD or TOA to determine the position. However, since the Earth itself is more similar to a rotating ellipsoid, an error between the result and the real value will be generated when using the sphere, and the error will be more significant in high-latitude areas when the sphere is established by the semi-major axis [22]. To reduce the error due to the projection to the approximate sphere, this paper improves the existing positioning algorithm by using Equation (Equation 42) and incorporating Equation (Equation 51) to derive the ellipsoidal pseudorange positioning (EPP) algorithm. This is relatively complicated, because the higher-order terms are not eliminated. Moreover, Equation (Equation 46) is used to derive the summation. Still, the ellipsoid is skipped at the same time, and the result of using the sphere to solve for the ellipsoid is more complicated than the real value. However, at the same time, due to skipping the ellipsoid–spherical–ellipsoid coordinate transformation and directly simplifying the formula for geodesic line length calculation on the ellipsoid, the singular phenomenon can be avoided in part of the region, and the applicability of the algorithm can be improved.

By setting a smaller threshold ε, when ΔB≤ε and ΔL≤ε, the positioning result is output; otherwise, the above calculation process is repeated. The flowchart is shown in Figure 3.

## 3. Theory of Cross-Station Chain Localization Based on Extended Kalman Filter

Due to the complex terrain in inland areas, it is often impossible to receive Loran-C signals from a group of station chains simultaneously, and the signals are frequently received from several stations of different station chains. This greatly limits the application scope of Loran-C. For example, in the Jiugongshan area of Hubei Province, the Loran-C signals are received with the UN-152B receiver, UrsaNal Company, Corporate Headquarters 616 Innovation DriveChesapeake, VA 23320, United States as shown in the following Figure 4.

The Loran-C signals received here, from the 7430 station chain and the 6780 station chain, are incomplete. The 8390 station chain can be obtained in full, but the SNR of the signals from the 8390X station is only 0.11. Due to the poor signal quality, the 8390 station chain cannot be used for positioning. In this conext, pseudorange localization methods demonstrate better applicability, and the simultaneous reception of signals from three stations can facilitate localization. The signals of 8390M, 8390Y, and 6780M can be used for pseudorange positioning.

The transmission delay within the same group of station chains is known (the transmission delay of Loran-C station chains in China is not disclosed), and cross-station chain positioning also requires the transmission delay of the two station chains, which is not stipulated in the Loran-C system. External GNSS timing can be used to easily measure the transmit delays of the two station chains, which defeats the purpose of using Loran-C as a GNSS backup system, since the Loran-C positioning solution is a nonlinear process. In this paper, we will utilize the EKF method to estimate the transmission delays of the 8390 station chain and the 6780 station chain and then perform cross-station chain positioning.

### 3.1. EKF Introduction

The mathematical framework of the Extended Kalman Filter [23] as a core algorithm for the state estimation of nonlinear systems can be expressed as follows. Nonlinear state space equations: $x_{k}=f\left(x_{k-1},u_{k-1}\right)+w_{k-1}\;z_{k}=h\left(x_{k}\right)+v_{k}$.

Included among these, $x_{k}\in R^{n}$ is the state vector of the system at the moment *k*; f(·) and h(·) are nonlinear state transfer and observation functions, respectively; $w_{k}∼N\left(0,Q_{k}\right)$ and $v_{k}∼N\left(0,R_{k}\right)$ are uncorrelated process noise and observation noise.

Local linearization is achieved by a first-order Taylor expansion:(52)$$\begin{array}{cc} F_{k} & =\left.\frac{𝜕f}{𝜕x}\right|\hat{x}k|k-1 \end{array}$$(53)$$\begin{array}{cc} H_{k} & =\left.\frac{𝜕h}{𝜕x}\right|\hat{x}k|k-1. \end{array}$$

The EKF consists of two steps, namely the prediction step(54)$$\begin{array}{cc} \hat{x}k|k-1 & =f(\hat{x}k-1|k-1,u_{k-1}) \end{array}$$(55)$$\begin{array}{cc} P_{k|k-1} & =F_{k}P_{k-1|k-1}F_{k}^{T}+Q_{k} \end{array}$$
and the update step(56)$$\begin{array}{cc} K_{k} & =P_{k|k-1}H_{k}^{T}{\left(H_{k}P_{k|k-1}H_{k}^{T}+R_{k}\right)}^{-1} \end{array}$$(57)$$\begin{array}{cc} \hat{x}k|k & =\hat{x}k|k-1+K_{k}\left(z_{k}-h\left(\hat{x}k|k-1\right)\right) \end{array}$$(58)$$\begin{array}{cc} Pk|k & =\left(I-K_{k}H_{k}\right)P_{k|k-1} \end{array}$$

### 3.2. Application of EKF in Cross-Desk Chain Time Synchronization

The first step is to initialize the parameters; we then estimate the current position, velocity, receiver clock difference, and cross-station chain clock difference and obtain the 6-dimensional observation matrix $\left[B_{0},L_{0},V_{N},V_{E},Dt_{r},Dt_{c}\right]$, where $B_{0}$ is the latitude probability, $L_{0}$ is the longitude probability, $V_{N}$ is the estimated velocity in the north direction, $V_{E}$ is the calculated velocity in the east direction, $Dt_{r}$ is the receiver clock difference, and $Dt_{c}$ is the cross-station chain clock difference. These are the key data that need to be computed by the EKF.

We set the state covariance matrix P, the process noise covariance matrix Q, and the state transfer matrix F; obtain the pseudorange observations of the three stations; and perform the EKF computation of $Dt_{c}$ according to Figure 5. After completing the cross-station chain clock difference calculation, the pseudorange localization calculation is carried out according to Section 2.6 to complete the cross-station chain pseudorange localization. The EKF process is shown in Figure 5.

## 4. Simulation and Testing

### 4.1. Theoretical Calculation of GDOP Values near Each Loran-C Station Chain

The ellipsoidal pseudorange positioning (EPP) algorithm has been derived in the previous section. The following simulation analyses are conducted to evaluate the positioning performance of this algorithm and the traditional single-station chain spherical hyperbolic positioning (SHP) algorithm. The GDOP [24] values of the two algorithms are calculated separately. Due to the different coverage areas of each chain in China, the positioning results in other regions need to be analyzed. In the following, the GDOP is calculated for the regions near the 7430 station chain, 8390 station chain, and 6780 station chain in China.

The geometric dilution of precision (GDOP) is a commonly used index in navigation, reflecting the geometric configuration of the receiver and signal stations regarding the positioning error, which is influenced by the integrated amplification effect of the index. The smaller the value, the higher the positioning accuracy.

The simulation calculates the positioning results of different station chains of the Loran system, assuming that the bias due to the signal propagation environment is negligible. Specifically, it is believed that the Loran signal TOA consists of PF only at this time. We select *c* = 299,792,458 m/s, η=1.000338. We calculate the theoretical GDOP values for each chain. Figure 6 shows a comparison of the GDOP values for the two algorithms for the 7430 chain; Figure 7 shows a comparison of the GDOP values for the two algorithms for the 8930 chain; Figure 8 shows a comparison of the GDOP values for the two algorithms for the 6780 chain. The intersection of the white dashed lines in each figure is the Loran-C navigation station.

Comparing Figure 6, Figure 7 and Figure 8, it is evident from the GDOP distribution map that the overall GDOP values near each chain when using the EPP algorithm are relatively low, while the GDOP values near each chain when using the SHP algorithm exhibit a star-shaped distribution based on the geometric layout of the chains. We calculate the ratio of the usable positioning areas for the SHP algorithm and the EPP algorithm, taking GDOP≤5. The result for the 7430 station chain is 43.64%; for the 8390 station chain, it is 26.00%; and, for the 6780 station chain, it is 26.04%. There are differences depending on the geometric position of the station chain. It can be concluded that the scope of application of the EPP algorithm is significantly greater than that of the SHP algorithm, as shown in Figure 9.

### 4.2. Comparison of Measured Data

This experiment collected data in Maoming, Guangdong Province, China. There are no differential stations in the local area, so SF+ASF was calculated using the theory in Section 2.1, Section 2.2 and Section 2.3. The terrain of the Maoming area is shown in Figure 10. Most of Guangdong Province and Guangxi Province consists of general mountainous areas with altitudes below 500 m. The average terrestrial geomagnetic parameters, relative permittivity ε=20, and terrestrial conductivity $\Sigma =3\times 10^{-3}$ were selected from Table 1.

The UN-152B receiver produced by Ursa Navigation Solutions, Inc. was used to receive Loran-C signals and measure the delay information from the receiving point to each station. The built-in delay of the receiver is 72.6 μs. UN-152B is a mature Loran-C receiver that is capable of some algorithmic suppression of receiver clock errors and multipath errors. On the day of the experiment, there was light rain. A vehicle-mounted receiver and computer were used, with the antenna placed outdoors in an open area. A total of 100 sets of TOA observation data with signal strength (SNR) greater than 5 dB were selected. Using a MATLAB program, the average positioning results were calculated, and the two methods, EPP and SHP, were used to compute the positioning values. The UN-152B measurement data were grouped, as shown in Figure 11. The UN-15B receiver and antenna are illustrated in Figure 12.

Table 2 shows the average values of the time delay information between the measured points and each station. SF+ASF is calculated as in Section 2.1, Section 2.2 and Section 2.3.

Table 3 shows the measured GNSS points and the EPP algorithm and SHP algorithm calculation values, and the signal processing flow is shown in Figure 13.

Figure 14 shows a comparison of the positioning errors between EPP and SHP based on GNSS positioning. As can be seen from the figure, EPP significantly reduces the positioning errors. After data processing, the RMSE for EPP and SHP is as shown in Figure 13.

As shown in Figure 15, the RMSE of EPP is 89.7% higher than that of SHP, indicating that the EEP algorithm can effectively improve the positioning accuracy of Loran-C.

### 4.3. Cross-Station Chain Pseudorange Localization Experiment

The measured data of Jiugong Mountain in Hubei Province are selected for pseudorange localization calculation, and the GNSS antenna coordinates are $29.39287^{\circ }$ N, $114.652715^{\circ }$ E. We select the relative dielectric constant ε=20 and the Earth conductivity $\Sigma =3\times 10^{-3}$, and the measured data are processed by MATLAB, as shown in Figure 4. The data of 6780M, 8390M, and 8390Y are selected for the localization process. The cross-station chain clock difference filtered image is shown in Figure 16. Taking the GNSS localization as the reference value, the EPP localization error is as shown in Figure 17, and the RMSE is calculated to be 78.4 m. Regarding the reason for the increase in error in this experiment, this comprises two aspects. One is the estimation aspect of the cross-station chain time difference, which remains difficult to estimate accurately, even with the EKF. The second reason is that Jiugongshan is very far away from the Loran-C stations, and the average value of the geoelectric parameter is impractical when estimating SF+ASF, which leads to a significant localization bias.

### 4.4. Discussion of Applicability of EPP Algorithm in Inland Regions

Inland regions typically feature complex terrain, including marshes, mountains, valleys, and forests, which are intertwined and complex. Additionally, these areas lack eLoran station support, resulting in significant positioning errors with traditional Loran-C systems. In such cases, the EPP algorithm, as outlined in Figure 13, significantly improves the positioning accuracy. This experiment was conducted in Maoming City, Guangdong Province, China, where the terrain is mountainous, and mountains, rivers, and dense forests characterize the area within the 6780 station chain. According to the GDOP value predictions in Figure 13, the GDOP value of the EPP algorithm at the test site is significantly lower than that of the SHP algorithm. Since the EPP algorithm achieves a transition from spherical positioning to ellipsoidal positioning, and pseudorange positioning has algorithmic advantages over hyperbolic intersection positioning, the EPP algorithm can provide higher positioning accuracy in inland areas with complex terrain.

## 5. Conclusions

This paper derives the Loran-C ellipsoidal pseudorange positioning formula and the EPP algorithm and compares and analyzes the GDOP distribution, RMSE, and maximum error of the EPP algorithm and the SHP algorithm. The simulation results show that the positioning area of the EPP algorithm is 129.1% to 284.6% greater than that of the SHP algorithm. Experiments show that the RMSE of the EPP algorithm is 89.7% lower than that of the SHP algorithm. After using the EKF to estimate the cross-station chain clock difference, cross-station chain pseudorange localization can be carried out to a certain extent, and the application scope of Loran-C can be significantly improved. Through the theoretical derivation, simulation calculations, and field experiments presented in this paper, it is shown that the proposed EPP algorithm can effectively improve the positioning accuracy and applicable area of Loran-C.

## Figures and Tables

**Figure 1 sensors-25-05110-f001:**
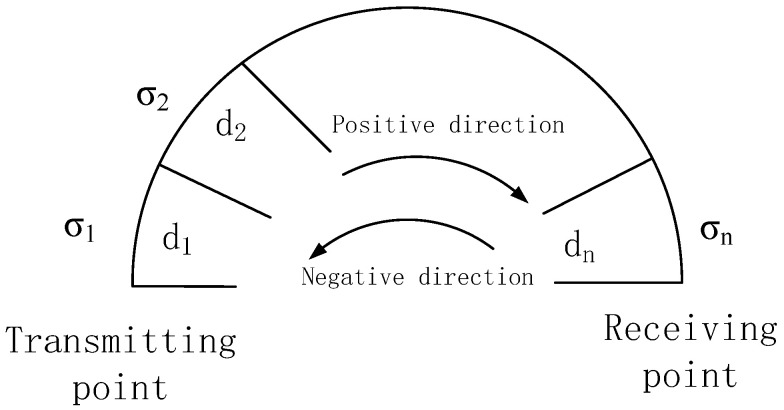
Millington formula schematic diagram.

**Figure 2 sensors-25-05110-f002:**
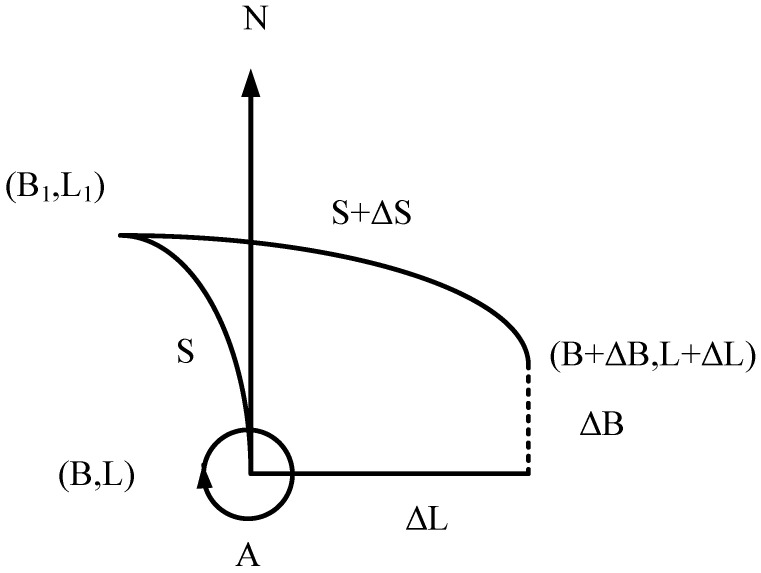
The relationship between the change in the geodesic line length and the change in the endpoint on the rotating ellipsoid.

**Figure 3 sensors-25-05110-f003:**
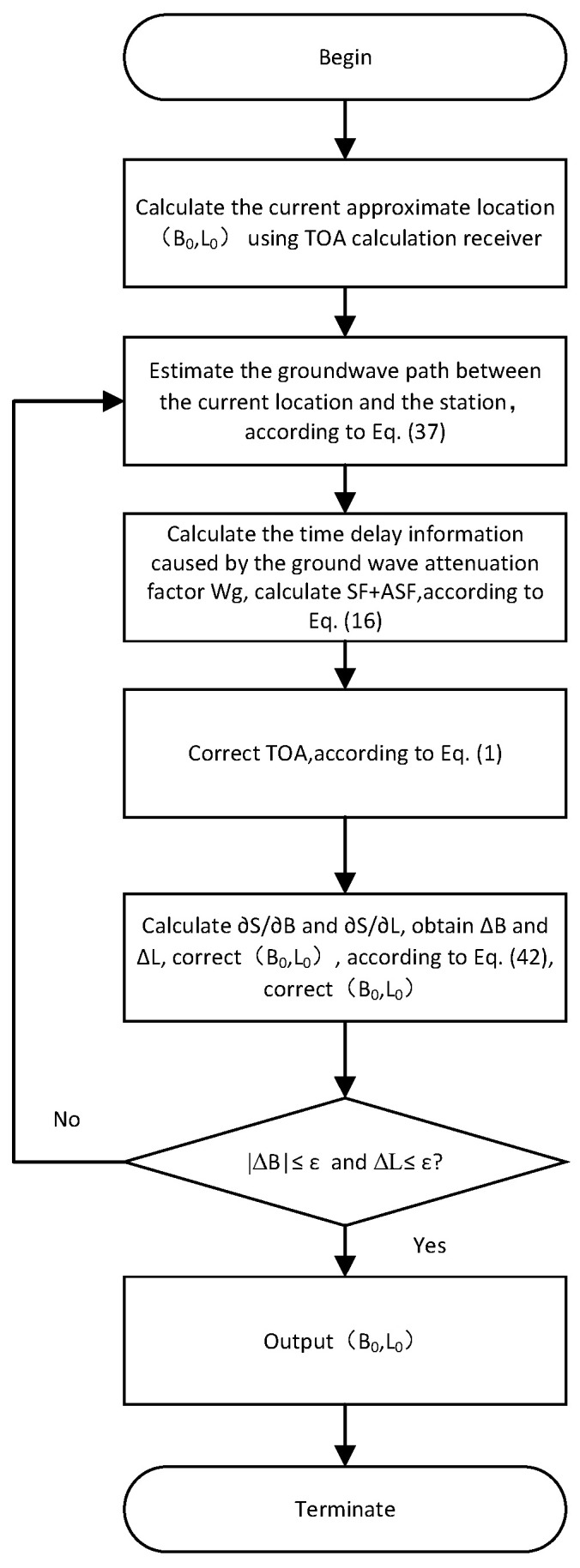
Pseudorange positioning logic flowchart.

**Figure 4 sensors-25-05110-f004:**
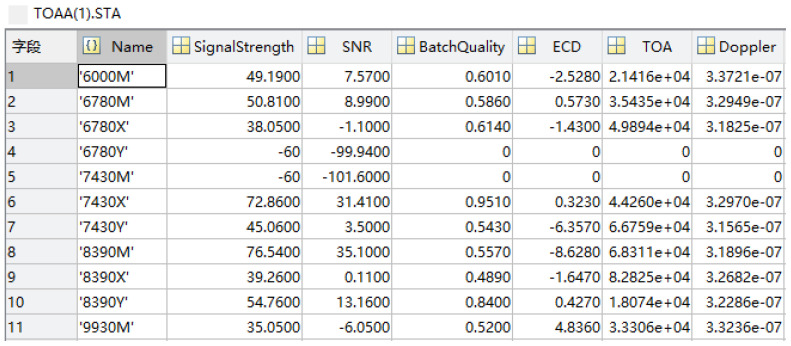
Screenshot of measured data from Jiugong Mountain, Hubei Province.

**Figure 5 sensors-25-05110-f005:**
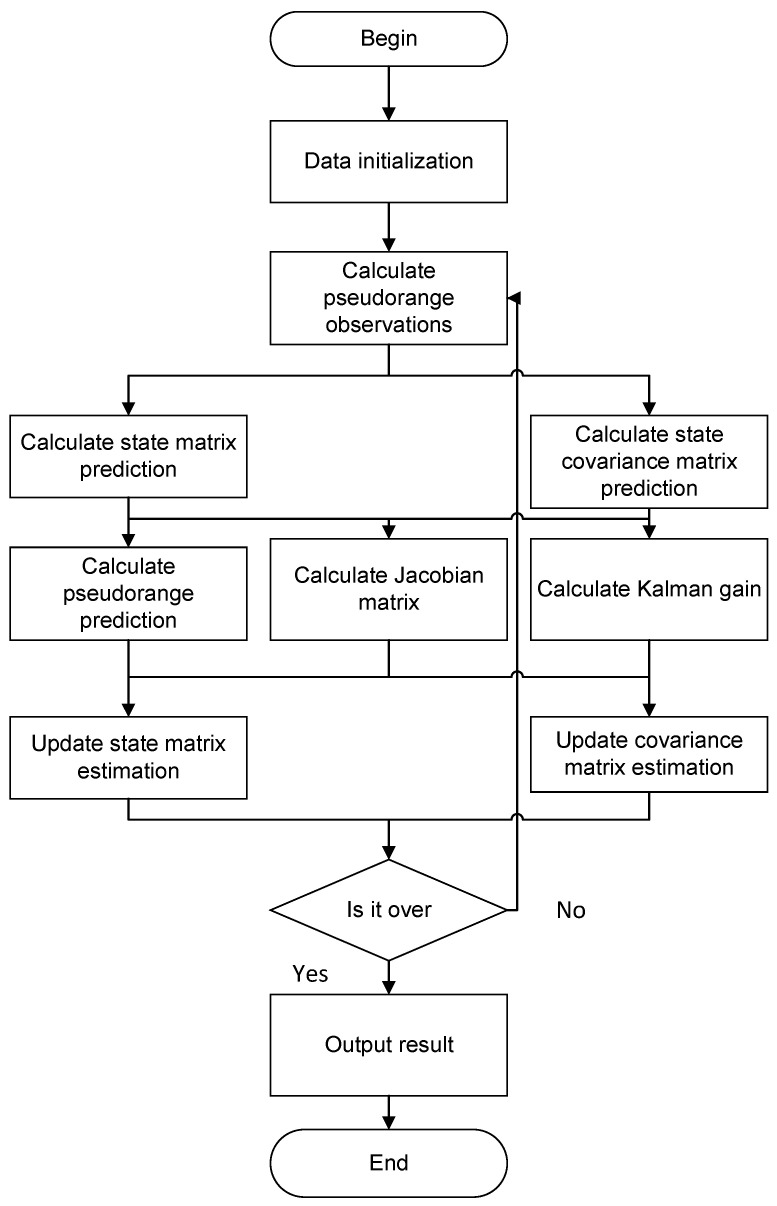
Flowchart of EKF-based cross-station chain clock difference estimation.

**Figure 6 sensors-25-05110-f006:**
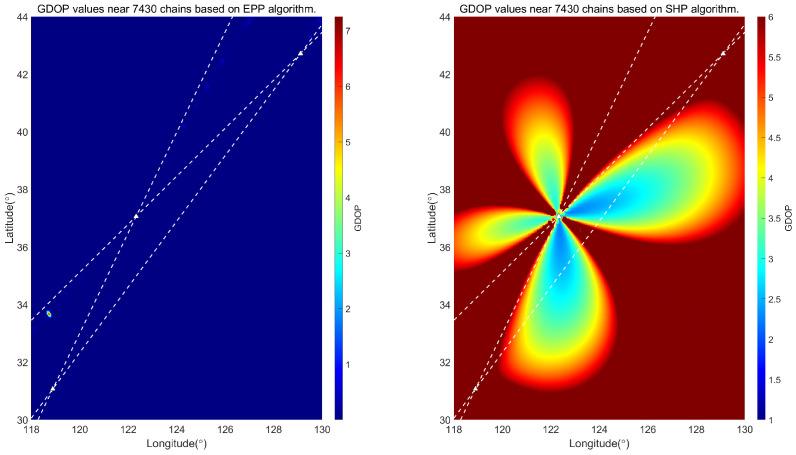
Comparison between EPP algorithm and SHP algorithm in locating GDOP value near 7430 station chain.

**Figure 7 sensors-25-05110-f007:**
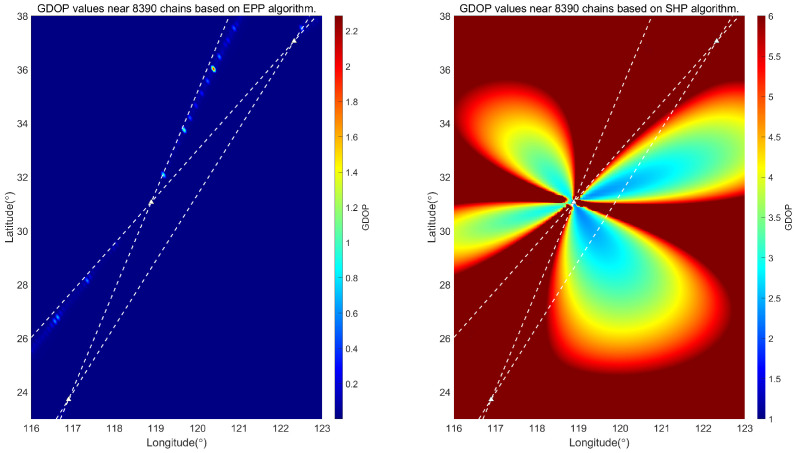
Comparison between EPP algorithm and SHP algorithm in locating GDOP value near 8390 station chain.

**Figure 8 sensors-25-05110-f008:**
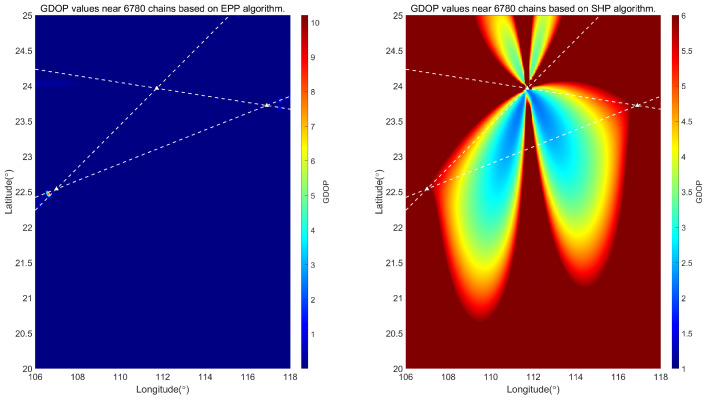
Comparison between EPP algorithm and SHP algorithm in locating GDOP value near 6780 station chain.

**Figure 9 sensors-25-05110-f009:**
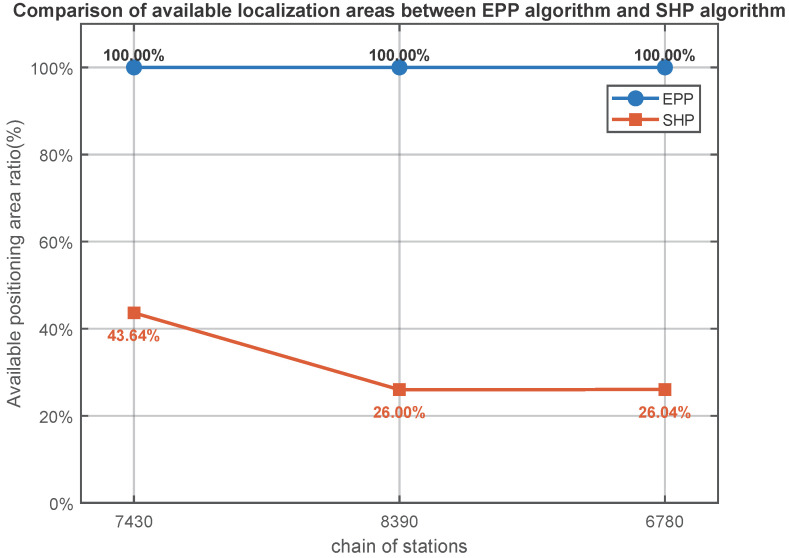
Comparison of available localization areas between EPP algorithm and SHP algorithm.

**Figure 10 sensors-25-05110-f010:**
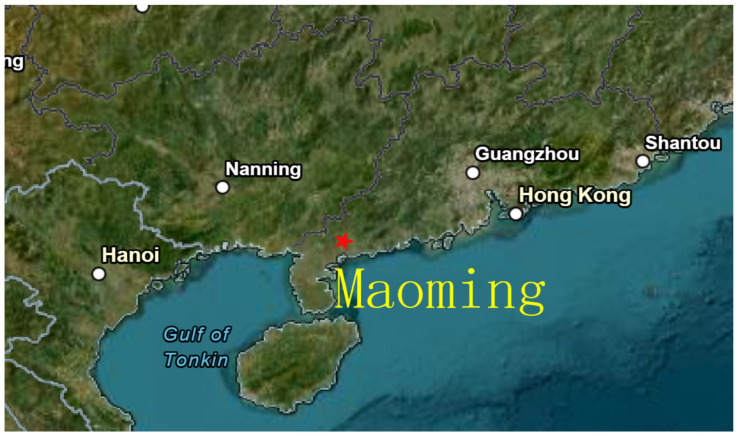
Satellite map of South China region.

**Figure 11 sensors-25-05110-f011:**
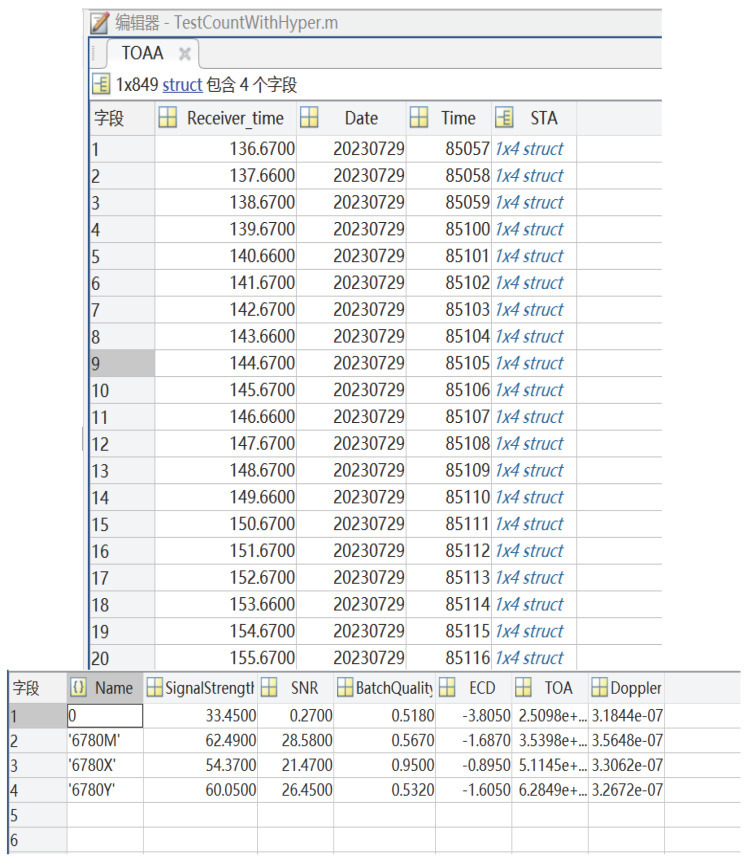
Actual measurement data extraction.

**Figure 12 sensors-25-05110-f012:**
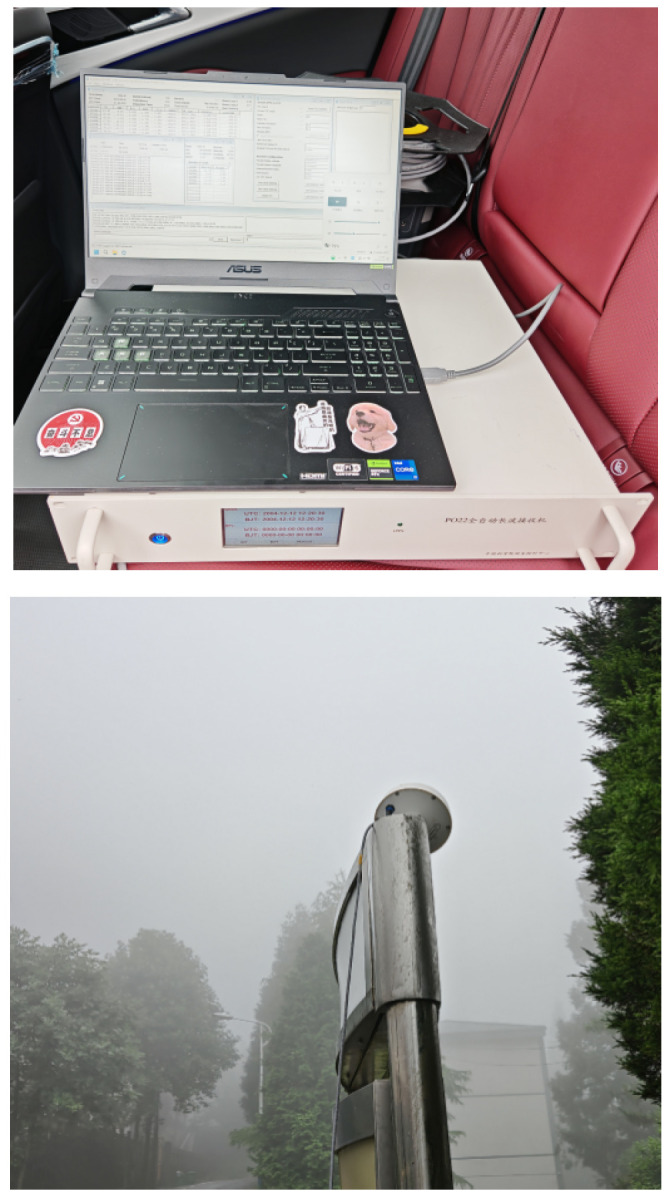
Experimental equipment.

**Figure 13 sensors-25-05110-f013:**
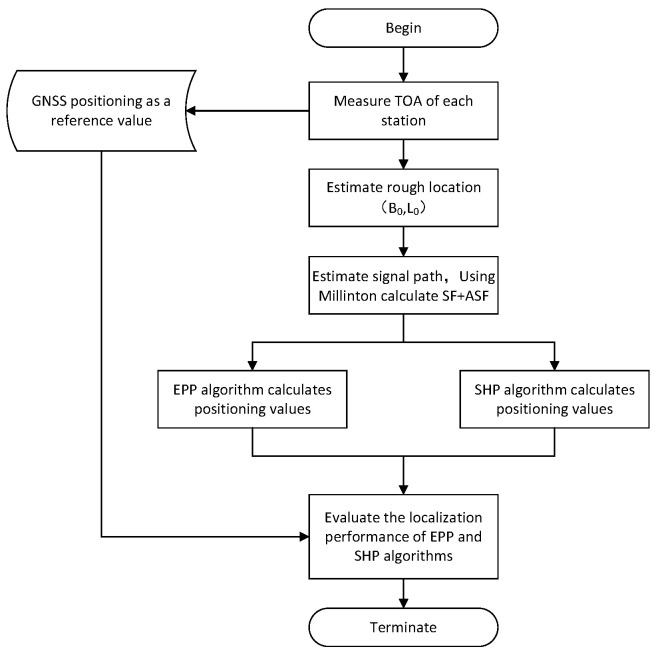
Signal processing flowchart.

**Figure 14 sensors-25-05110-f014:**
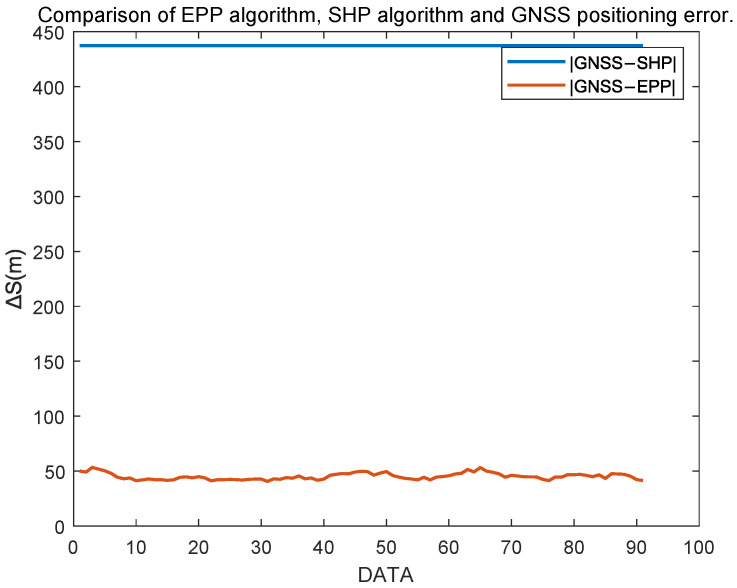
Comparison of EPP and SHP positioning results.

**Figure 15 sensors-25-05110-f015:**
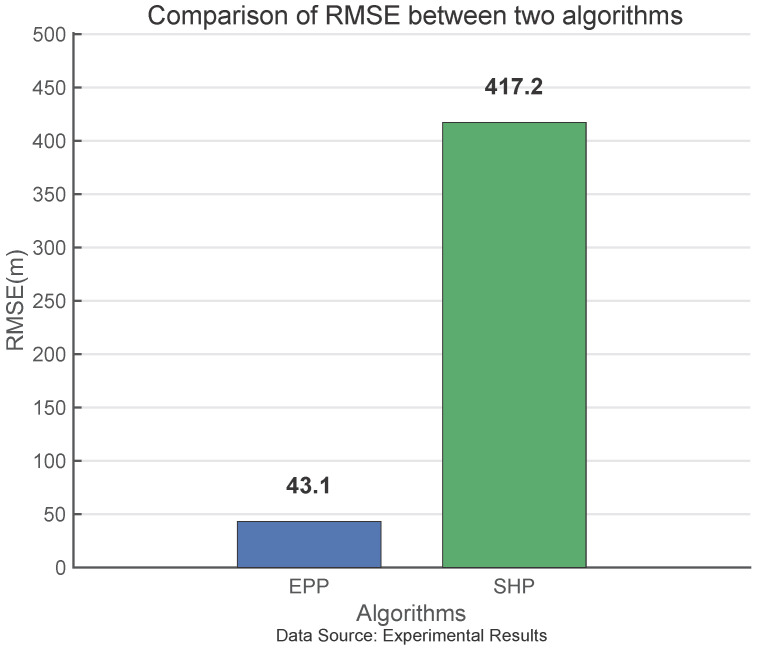
Comparison of RMSE between two algorithms.

**Figure 16 sensors-25-05110-f016:**
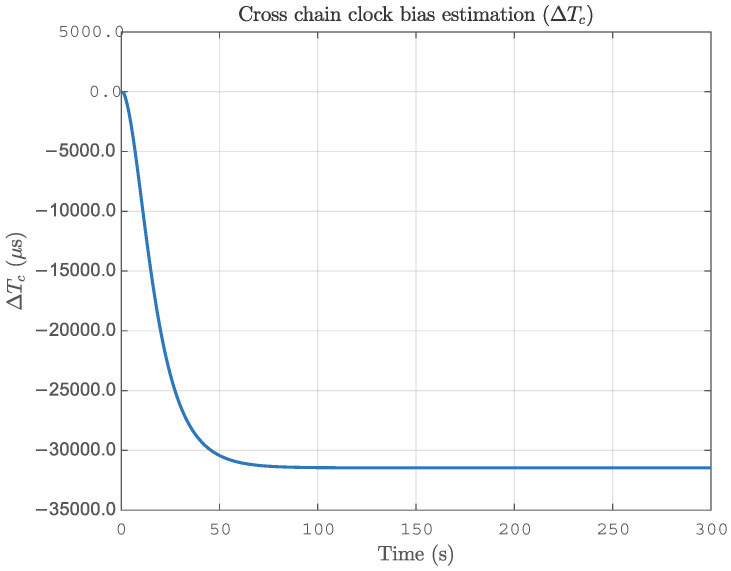
Cross-station chain clock difference filtered image.

**Figure 17 sensors-25-05110-f017:**
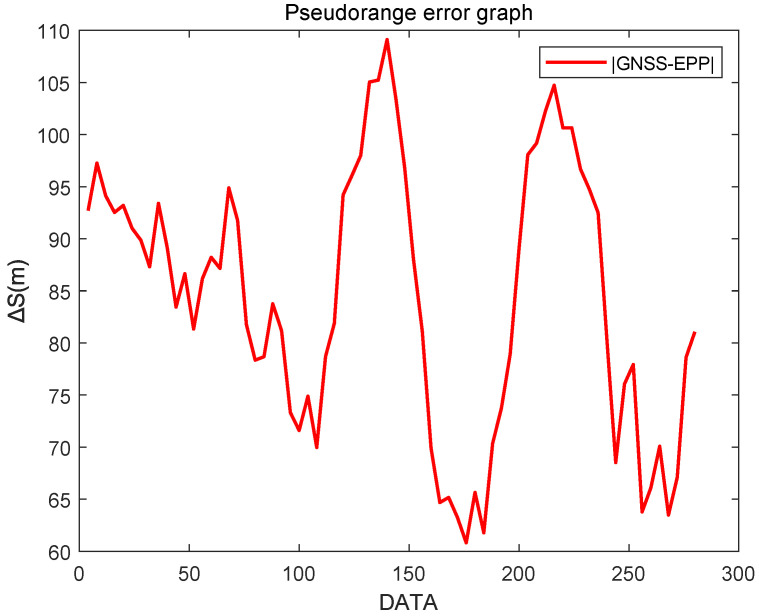
Cross-station chain pseudorange localization error image.

**Table 1 sensors-25-05110-t001:** Four typical ground electrical parameter pairings.

Serial Number	Ground Type	Relative Permittivity ε	Nominal Conductivity Value Σ (S/m)
1	Good electrical conductivity	40	$3\times 10^{-2}$
2	Damp ground	30	$1\times 10^{-2}$
3	Average land	22	$3\times 10^{-3}$
4	Drier land	15	$1\times 10^{-3}$

**Table 2 sensors-25-05110-t002:** Time delay values between measured points and various stations.

Delay Value	6780M	6780X	6780Y
TOA (μs)	35,398.06	51,144.61	62,848.65
ME (μs)	0.43	1.23	0.74

**Table 3 sensors-25-05110-t003:** Maoming measured data.

GNSS	EPP	SHP
$L_{G}$ 111.08667 $B_{G}$ 21.78716	$L_{E}$ 111.08705 $B_{E}$ 21.78716	$L_{S}$ 111.08266 $B_{S}$ 21.78721

## Data Availability

Dataset available on request from the authors.

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
