# Peer review of "Research on the Loran-C Pseudorange Positioning Method Based on an Ellipsoidal Geodesic Model and Its Application in Inland Areas"

_sensors, 2025, doi:10.3390/s25165110_

Round 1
Reviewer 1 Report
Comments and Suggestions for Authors
The manuscript presents a relevant and original contribution by deriving an ellipsoidal pseudorange positioning (EPP) method for the Loran-C system, specifically targeting its limitations in inland areas. The study is comprehensive, combining mathematical derivation, simulation, and real-world experiments. The results demonstrate the superiority of the EPP algorithm over the conventional spherical hyperbola positioning (SHP) method, both in simulation (with an increase of up to 284.6% in coverage area) and field experiments (reducing RMSE by 89.7%). The discussion on the applicability of the algorithm in complex terrain is particularly valuable.
Suggestions for improvement:
1) Although the methodology is well described, it would be beneficial to add a more detailed discussion of the limitations and possible challenges in extending the method to other environments or different Loran-C configurations.
2) The authors could discuss potential integration with GNSS for redundancy in navigation and timing applications, given the current context of resilient PNT systems.
3) Consider summarizing the main practical recommendations for implementation at the end of the conclusions section.
Author Response
Thank you for the comments. I have encountered many issues in my previous paper writing, and their suggestions are very valuable. I have accepted all of them and made the necessary corrections to the article. Below are the specific corrections.
Regarding Reviewer 1's comments:
Thank you for the valuable comments from the reviewer. I have accepted them all and made the necessary corrections.
For opinions 1 and 2, the article has added research on cross link pseudorange positioning based on EKF, focusing on the application value of Loran-C as a GNSS backup system, mainly in the last three paragraphs of the first part, the third part, and the third section of the fourth part.
Regarding Opinion 3, the article has strengthened the discussion on the application of Loran-C pseudorange positioning in the summary section, mainly focusing on the fifth part.
Reviewer 2 Report
Comments and Suggestions for Authors
The study addresses an inaccuracy of Spherical Hyperbola Positioning (SHP) in inland areas due to spherical Earth assumptions, which is a well-defined problem in Loran-C positioning. The authors propose an Ellipsoidal Pseudorange Positioning (EPP) method, which can be considered a logical and meaningful innovation that addresses the GDOP issue and enhances positioning reliability in inland areas. However, despite the paper's structure adhering to academic norms, there are some areas for improvement in the overall text content.
The reviewer’s comments, questions, and suggestions for improving the manuscript are given below:
- The manuscript would benefit from a critical analysis of its contribution in comparison to state-of-the-art techniques in GNSS error correction, positioning fusion methods, or Kalman filtering.
- The manuscript lacks a deeper comparative analysis with other modern alternatives, such as advanced GNSS corrections or hybrid systems. The EPP methods are not well presented, and it is more like a derivative adaptation of the GNSS concept than a distinct innovative approach.
- In some sections (e.g., Introduction, Methodology), there is significant redundancy and headings are inconsistent (e.g., ‘Geodesic Theme Issues in eLoran System Positional Solving’ in non-standard). There is no smooth transition from concepts to theoretical conclusions and empirical validation, which makes the text disjointed and inconsistent.
- The technical description of the method and mathematical procedures is comprehensive. However, the style of places should be edited to be more transparent and easier to understand. Here are some examples: “The Loran-C system uses the spherical hyperbolic positioning (SHP) method”, “Due to the low signal-to-noise ratio at some of the Loran signal reception points…”. Specific terminology is used that is inconsistent and interchangeable, which is misleading. For example, the terms “geodetic,” “geodesic,” “elliptical,” and “ellipsoidal” are sometimes used interchangeably, which can be misleading.
- The approach proposed by the authors is mathematically rigorous, and the modeling of real-world signal propagation, including ASF and SF delays, is commendable. The mathematical presentation of the method would benefit from an explanation of all symbols used, as well as a good combination of superscript and subscript.
- The suggested approach is tested in the real world in Maoming, Guangdong—a region characterized by complex terrain. The manuscript would benefit from more detailed experimental tests regarding the setup, which lacks control variables (hardware-induced delays, multipath interference), and the use of a single receiver model (UN-152B) without discussing calibration, clock bias handling, or error margins. These circumstances raise questions about the generalizability of the suggested approach and its applicability to other geographical regions.
- The discussion section should comment on the method’s general effectiveness in terms of scalability limits, environmental sensitivities, and integration challenges within existing Loran-C infrastructure worldwide, as well as at regional and/or national levels.
General recommendations:
- Тo improve the structure of the presentation and the transitions from theoretical to empirical sections.
- It is advisable to expand the experimental part by using different configurations of the devices.
- To improve the literature review by comparing and highlighting the advantages and limitations of the EPP method proposed by the authors
The manuscript is to be carefully revised for English, ideally by an editor whose native language is English.
Author Response
Thank you for the valuable comments . I have accepted them all and made the necessary corrections.
Regarding Opinion 1, the article has added research on cross link pseudorange positioning based on EKF, with a focus on discussing the application value of Loran-C as a GNSS backup system, mainly in the last three paragraphs of Part One and Part Three.
For Opinion 2, the article added experiments in the Jiugongshan area of Hubei Province and completed cross platform chain positioning in Section 4.3 of the article.
Regarding Opinion 3, the article discusses the reasons for the increase in cross platform chain positioning error in the experimental section of 4.3. At the same time, the article has been revised in terms of overall English expression.
Reviewer 3 Report
Comments and Suggestions for Authors
Author may want to check if certain wordings are to be replaced by a more formal word. For example, in page 2, “Literature [11][12] derives…” could be more appropriately as “References [11] and [12] derive…”. Author may let a professional to review the paper for or to utilize any grammar software to improve the article content. Because some of the content seems to be direct translation from another language. Some sentence seems to be not making sense as I could not able to connect it with previous sentence, such as in Section 2.1 and 2.2. Also, author to make sure the paragraph formatting is consistent. For example, Page 2, between line 83 and line 84.
- In Page 2 last paragraph of section 1, author may need to elaborate more as there seems to be missing connection to section 2.
- Section 2.1, Page 3, Equation (1). What is the unit of TOA, PF, SF and ASF? It is good to indicate j is complex number.
- For Equation (2), what is R? Also, the beam angle of the ground wave attenuation factor “arg Wg” is expressed with respect to?
- For Equation (6), what is x and t2?
- In Page 6 Line 178/179, “a,b are the ellipsoid short and long semi-axes…”, the more correct terminology is “semimajor axis” and “semiminor axis”.
- Also, for Equation (19), (20) and onwards, there are some symbols overlapping with previous equations that have different meaning. Please use another representation.
- For Equation (20), please provide the reference/citation of Bessel differential equation.
- For Equation (20), what are delta sigma and delta lambda? (They should be defined immediately after this equation instead of several rows away).
- Equation (23) does not show any integration of Delta S? Can author show or proof how (20) – (22) leads to (23) then (24)?
- Also, for Equation (24) it shows “S”, but the sentence above says “Delta S”?
- Page 7 line 197. “…. Due to Eq. (2)”, but I could not confirm if Equation (2) is the correct one.
- Page 7, is “A0” a constant?
- Equation (32), what is “d” infront of “Delta”? In fact, can author show and proof the details process from Equation (32) to (35)?
- For Equation (38), Delta S is defined in another way, do we require the Delta S equation defined previously? If not, why it exist?
- In Page 9, geodesic or geodetic? As both has different meanings.
- In page 9, what is ellipsoid long half-axis length?
- For figure 3, it will be better if equation numbers are included into it.
- In Page 12, how author obtain the value of GDOP? Or how author calculates it?
- Page 19 TOA is maximum error?
Author Response
Thank you for the valuable comments. I have accepted them all and made the necessary corrections.
Regarding Opinion 1, it has been corrected by adding an introduction to the overall layout of the article after the first part.
Regarding Opinion 2, it has been annotated on page 2 of the PDF.
For Opinion 3, the "R" symbol you proposed does not appear in Formula 2. "arg Wg" is the radiation angle of the ground wave attenuation factor, which is introduced in Formula 6.
For Opinion 4, x is an intermediate variable and t2 is a typo, which has been corrected to ts.
Regarding Opinion 5, it has been corrected.
For Opinion 6, the symbols have been corrected throughout the text, and the earth conductivity has been replaced with the symbol Sigma.
For opinions 7-10, these contents are from reference [20], which is a mature mathematical formula.
Regarding Opinion 11, the typo has been corrected.
For opinion 12, A0 is an unknown variable.
For opinions 13-14, this formula is also from reference [20]. It is a mature mathematical formula, and the specific derivation process is extremely complicated. I have briefly described the derivation process in the article. Sorry.
I have made corrections to the English expression throughout Opinion 15.
For Opinion 16, the value of the long half axis is 6378137.
For Opinion 17, Figure 3 has been modified by adding equation numbers.
For Opinion 18, an introduction to GDOP has been added on page 14 of PDF. GDOP is a commonly used indicator in the navigation field, and the calculation method is detailed in reference [24].
For Opinion 19, TOA is the signal propagation time.
Round 2
Reviewer 2 Report
Comments and Suggestions for Authors
The authors have reflected all the comments from the review, supplementing the study and making the appropriate corrections and additions to the text.
A small comment on citation. Literature sources are cited in the manuscript by indicating them in parentheses by the numbers of the references. It is better names of the authors should be cited, followed by the corresponding publication number in parentheses, for example, on Line 58: "References [11] and [12] derive ..."; it is better to cite " Jinghu et al. [11] and Fengchang et al. [12] derive...". A similar citation is on Line 65.
Author Response
Comments1:[ It is better names of the authors should be cited, followed by the corresponding publication number in parentheses, for example, on Line 58: "References [11] and [12] derive ..."; it is better to cite " Jinghu et al. [11] and Fengchang et al. [12] derive...". A similar citation is on Line 65.]
Response:[The wording has been rewritten for references 12-14].
Reviewer 3 Report
Comments and Suggestions for Authors
see attachment

Author Response
Comments1:[Eq (2), the “R” (by arrow) has not been defined.]
Response:[Sorry, there is a typo in Eq.2.,R should be d. It has been corrected.]
Comments2:[. It is suggested to include reference before some equation, particularly (20), (32), (38).
For example in Page 7 Line 198, “Using the above conditions, derive the differential
formula for calculating arc length on an ellipsoid surface [20]:”, instead.]
Response:[Reference citations have been added for Eq. 20, 32, and 38]
Comments3[Page 23 Line 402, the definition is incorrect for TOA.]
Response:[The definition of TOA has been corrected on page 24 of the PDF.]